# Marine reserve benefits and recreational fishing yields: The winners and the losers

Mohsen Kayal[1,2,3]*, Marine Cigala[1,2], Eléonore Cambra[1,2], Nelly Soulat[1,2¤a], Manon Mercader[1,2], Audrey Lebras[1,2], Pauline Ivanoff[1,2], Léa Sébési[1,2], Aurélie Lassus-Debat[1,2¤b], Virginie Hartmann[4], Mélissa Bradtke[1,2], Philippe Lenfant[1,2], Coraline Jabouin[5], Julien Dubreuil[1,2¤c], Dominique Pelletier[6], Manon Joguet[5¤d], Solène Le Mellionnec[1,2], Marion Brichet[7], Jean-Louis Binche[4], Jérôme Payrot[4], Gilles Saragoni[1,2], Romain Crec'hriou[1,2], Marion Verdoit-Jarraya[1,2]

**1** UPVD-CNRS, Centre de Formation et de Recherche sur les Environnements Méditerranéens, UMR 5110, Université de Perpignan Via Domitia, Perpignan, France, **2** Centre de Recherche sur les Ecosystèmes Marins (CREM), Port-Barcarès, France, **3** Institut de Recherche pour le Développement, UMR 9220 ENTROPIE, IRD, Université de la Réunion, IFREMER, CNRS, Université de la Nouvelle-Calédonie, Nouméa, New Caledonia, **4** Département des Pyrénées-Orientales, Réserve Naturelle Marine de Cerbère-Banyuls, Banyuls-sur-Mer, France, **5** Parc naturel marin du golfe du Lion, Argelès-sur-Mer, France, **6** EMH, Institut Français de Recherche pour l'Exploration de la Mer (IFREMER), Nantes, France, **7** Direction Interrégionale de la Mer Méditerranée, Marseille, France

¤a Current address: SEANEO, Perpignan, France
¤b Current address: Parc naturel régional de Camargue, Arles, France
¤c Current address: Comité Régional des Pêches Maritimes et des Elevages Marins de Bretagne, Rennes, France
¤d Current address: FROM Nord, Boulogne-sur-Mer, France
* mohsen.kayal@ird.fr

## Abstract

Marine reserves constitute effective tools for preserving fish stocks and associated human benefits. However, not all reserves perform equally, and predicting the response of marine communities to management actions in the long run is challenging. Our decadal-scale survey of recreational fishing yields at France's 45-year old Cerbère-Banyuls marine reserve indicated significant protection benefits, with 40–50% higher fishing yields per unit effort in the partial-protection zone of the reserve (where fishing is permitted but at a lower level) than in surrounding non-reserve areas. Over the period 2005–2014, catch per unit effort (CPUE) declined both inside and outside the reserve, while weight per unit effort (WPUE) increased by 131% inside and decreased by 60% outside. Different CPUE and WPUE trajectories among fish families indicated changing catch assemblages, with yields increasing for the family most valued by fisheries, Sparidae (the ecological winners). However, reserve benefits were restricted to off-shore fishermen (the social winners), as on-shore yields were ~4 times lower and declining, even inside the reserve. Our study illustrates how surveys of recreational fishing yields can help evaluate the effectiveness of marine protected areas for key social and ecological protagonists. We show that, more than four decades after its establishment, fishing efficiencies at the historical Cerbère-Banyuls marine reserve are still changing, but benefits in terms of catch abundance, weight, and composition remain predominantly restricted

**Data Availability Statement:** Data used in this study are available at https://doi.org/10.5281/zenodo.4149014.

**Funding:** This study was funded by multiple sources including the French Agency for Biodiversity (https://ofb.gouv.fr) and Region Occitanie/Pyrénées-Méditerranée (www.laregion.fr), Département des Pyrénées-Orientales (www.ledepartement66.fr), French Ministry of Ecology, Energy, Sustainable Development and Regional Planning (www.ecologique-solidaire.gouv.fr), European Fisheries Fund (https://ec.europa.eu), Pays Pyrénées Méditerranée (www.payspyreneesmediterranee.org), as well as the Regional Direction for the Environment, Planning and Housing Occitanie (www.occitanie.developpement-durable.gouv.fr). Data were collected in part during the PAMPA project on Indicators of MPA Performance funded by the French Ministry of Ecology and the French MPA Agency. The funders had no role in study design, data collection and analysis, decision to publish, or preparation of the manuscript.

**Competing interests:** The authors have read the journal's policy and have the following competing interests: NS took on a position as a paid employee of SEANEO after the time of study. This does not alter our adherence to PLOS ONE policies on sharing data and materials. There are no patents, products in development or marketed products associated with this research to declare.

to off-shore fishermen. Further regulations appear necessary to guarantee that conservation strategies equitably benefit societal groups.

## Introduction

Despite increasing management efforts, the decline of fishing yields remains a global concern [1]. This is especially true in the Mediterranean Sea where human impacts to the marine environment are diverse, intense, and increasing [2]. Indeed, the Mediterranean is considered the most overfished marine basin on the planet and poses severe management challenges, as exploitation of marine resources in this nearly-enclosed sea is shared among 21 bordering countries whose economic development is tied to activities impinging on the marine environment [3–5]. While Mediterranean marine biota are threatened by a multitude of anthropogenic stressors including pollution and eutrophication, climate change, invasive species, marine transport, aquaculture, and tourism, the major historical and current drivers of declining biodiversity and productivity are habitat loss and fishing [2, 6].

Fish landings in the Mediterranean have been decreasing since the mid-1980's, despite expanding fishing efforts toward lower trophic levels and the deeper sea [1, 7, 8]. This decreasing catch rate has resulted in declines of commercial fishing activities. In contrast, recreational fishing has been on the rise, particularly along the European coast of the Mediterranean [3, 4, 9]. As in many other regions, the relative contributions of commercial versus recreational fishing to the local socio-economy and decline of fish stocks have yet to be quantified throughout the Mediterranean [8, 10–12], though recreational fishing is suspected to exert a strong and increasing pressure, particularly on highly targeted marine species [9, 13–16]. In the Mediterranean and elsewhere, strategies for preserving fish stocks consist primarily of regulating fishing efforts through gear restrictions (gear type and number), fishing yields through quotas (catch size and bag limits), and fishing areas and seasons through exclusion zones and marine reserves [6, 10]. However, the long term effectiveness of these management strategies for preserving species abundance and ecosystem services is difficult to predict [17, 18]. In the face of such uncertainties, the preservation of fisheries resources and associated socio-economic benefits poses serious regulatory challenges in terms of implementing appropriate measures for resource durability and equitable access [3, 19]. In this context, identifying social and ecological protagonists vulnerable to environmental decline can refine regulatory strategies and help define win-win, sustainable management for people and ecosystems [20–22].

We performed a decadal scale survey (2005–2014) of recreational fishing activities at the Cerbère-Banyuls marine reserve (Fig 1), one of the oldest marine protected areas in the Mediterranean, to evaluate the effectiveness of local management efforts in preserving fisheries resources. The survey consisted of ~1,500 on-site interviews with recreational fishermen fishing inside and outside the reserve, and recorded ~6,000 individual catches representing a total weight of ~1 ton for a fishing effort of ~5000 line-hours. Within the reserve, fishing is subject to restrictions (gear restrictions and bag and size limits, see section 2.1.) and only takes place in a buffer zone of partial protection surrounding the fully protected no-take area (Fig 1). In contrast, no restrictions apply outside of the reserve where fishing follows the French national regulation for the Mediterranean Sea. Therefore, we hypothesized that fishing yields would differ between the partial protection zone of the reserve, which benefits from the vicinity of the fully protected no-take area and undergoes restricted fishing pressure, as compared to surrounding no-reserve areas. Similarly, because reserve benefits are often not equally distributed in space and among species undergoing different fishing pressure [17, 23–26], we also

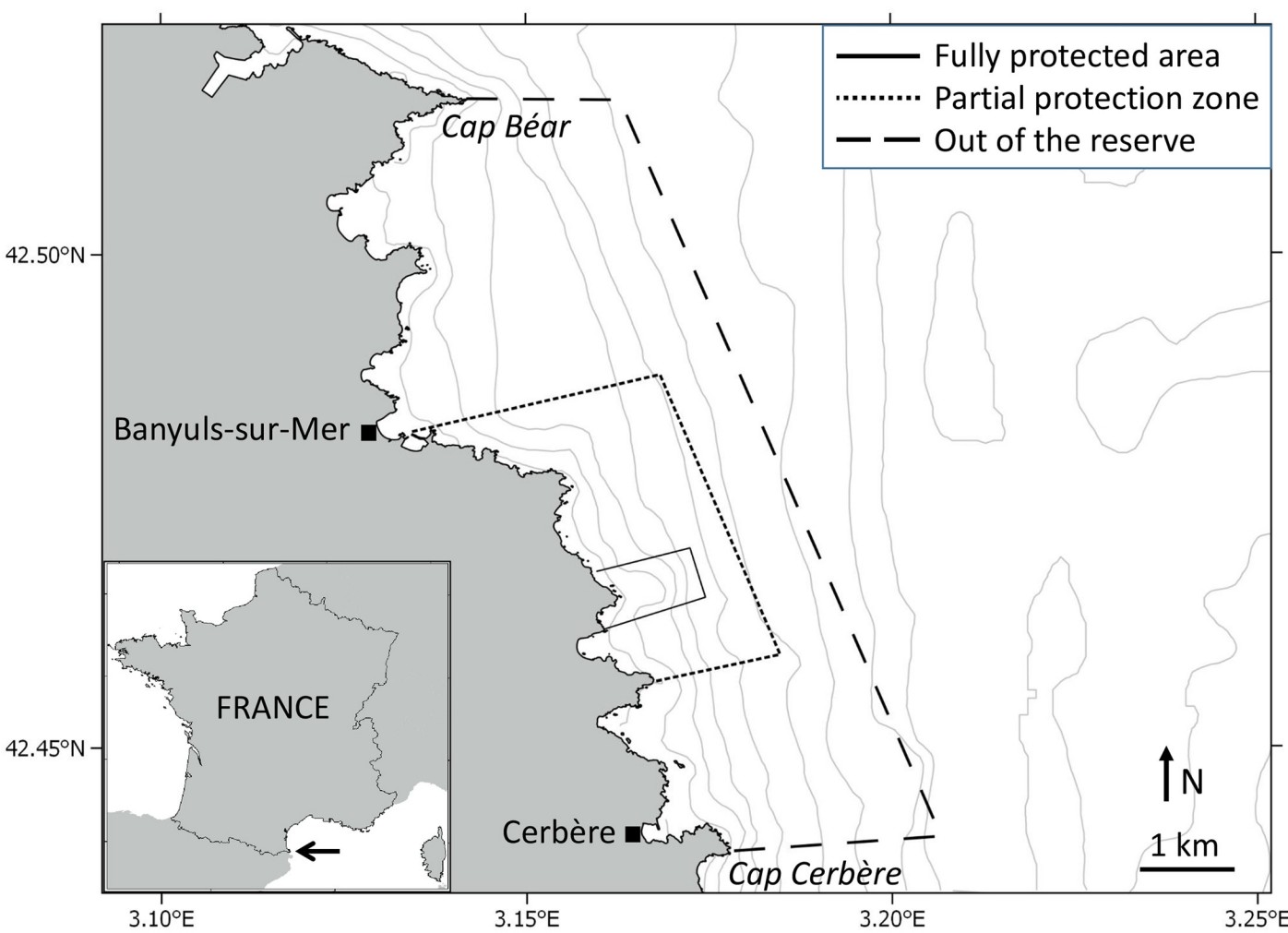

**Fig 1. Map of the study area indicating the position of the Cerbère-Banyuls marine reserve's fully protected core area (no-take zone), buffer zone of partial protection (fishing allowed with restrictions, see Methods), and control outside area (no specific regulation on fishing) in relation to the towns of Banyuls-sur-Mer and Cerbère and the two capes, Cap Béar and Cap Cerbère.** Established in 1974, Cerbère-Banyuls is one of the oldest marine protected areas of the Mediterranean Sea. The arrow in the insert indicates the position of the reserve in the natural marine park of the Gulf of Lion situated in the north-western corner of the Mediterranean, at the border between France and Spain. Isobaths indicate depth variation every 10 meters. Maps were produced using the open source program QGIS.

hypothesized that reserve benefits would differ for fishermen fishing on-shore (access limited to the coastline of the reserve) and off-shore (unrestricted boat access to the entire reserve), as well as among fish families differently targeted by fisheries.

We used three biological indicators commonly used to characterize fishing yields: catch abundance, weight, and composition. We tested for differences in catch per unit effort (CPUE) and weight per unit effort (WPUE) between fishermen fishing inside versus outside of the reserve, as well as on-shore along the coastline versus off-shore from boats. CPUE and WPUE trajectories were also compared among the three dominant fish families (S1 Table), namely Sparidae (sea breams), Serranidae (groupers), and Labridae (wrasses), each exhibiting different levels of species diversity, occupying different positions in habitats and trophic levels, and with different values for fisheries [3, 9, 27]. Based on an unprecedented survey of recreational fishing activities in the Mediterranean, our study provides a decadal-scale evaluation

of the effectiveness of the Cerbère-Banyuls nature reserve, one of the most preserved marine reserves in the region (see Methods), for supporting fishing yields. Our results shed light on the consequences of fishing regulations for key social and ecological protagonists with implications for adaptive management of fishery resources.

## Methods

### Ethics statement

This study involved interviews with recreational fishermen. The interviews were conducted anonymously after informed consent for study participation from each subject. The survey methodology and material was approved by the University of Perpignan.

### The Cerbère-Banyuls natural marine reserve

The Cerbère-Banyuls natural marine reserve (www.reserves-naturelles.org/cerbere-banyuls) is a French marine protected area situated in the north-western corner of the Mediterranean Sea (Fig 1). Established in 1974, it is one of the oldest marine reserves in the Mediterranean and is managed by the Departmental Council of the Pyrénées-Orientales. With over 100,000 annual visitors, including more than 30,000 scuba-divers in recent years, the Cerbère-Banyuls marine reserve contributes greatly to the region's community character and socio-economic development [11, 24]. Thanks to its ecological wealth and management, the reserve is recognized since 2014 as one of the 40 sites listed on IUCN's Green List of Protected Natural Areas (www.iucn.org), and is since 2018 among the 16 Blue Parks distinguished as outstanding marine protected areas by the Marine Conservation Institute (https://globaloceanrefuge.org). Since 2011, the reserve is part of the larger, 4,010 $km^2$ in area natural marine park of the Gulf of Lion (www.parc-marin-golfe-lion.fr). This is the largest marine park in the Mediterranean Sea, and is managed by the French Agency for Biodiversity. At this stage, there are no restrictions regarding fishing and other human usages in the park, though scientists, managers, and representatives regularly hold discussions through committees, workshops and ongoing projects to deliberate on future plans.

The reserve comprises a small nucleus of 65 ha of fully protected area, where only recreational navigation, surface swimming, and scientific diving are authorized (Fig 1). This area is surrounded by a larger, 650 ha buffer zone of partial protection where recreational activities such as scuba-diving, boating, and daytime angling are authorized, but subject to quotas and restrictions that are stricter than the general French coastal fishing regulations for the Mediterranean Sea which apply outside of the reserve [28–30]. During the studied period, anglers in the reserve were restricted to a maximum of 2 lines with up to 6 hooks if fishing on-shore, and 12 hooks if fishing from a boat. No restrictions on lines and hooks applied outside of the reserve. The number of recreational fishermen fishing in the reserve is regulated by a free but mandatory annual permit; up to 1,500 permits were issued annually over the course of this study. Species-specific minimum catch sizes and maximum bag limits also apply, and spearfishing is forbidden within the reserve [29]. Some commercial fishing also takes place in the partial protection zone of the reserve, with a fleet of 4–15 artisanal boats registered annually during the studied period [11]. Surveys of recreational and commercial fishing indicate that catches around the Cerbère-Banyuls marine reserve consist primarily of fish belonging to the families Sparidae, Serranidae, and Labridae [28, 30] (S1 Table). The reserve also hosts recovering populations of dusky grouper (*Epinephelus marginatus*, family Serranidae) and brown meagre (*Sciaena umbra*, family Sciaenidae) which are protected from fishing by national moratoria.

## Survey methodology and design

Recreational fishing activities in and around the Cerbère-Banyuls marine reserve were surveyed during four monitoring campaigns performed in 2005, 2009, 2010–2011, and 2013–2014 in an area expanding from Cap Béar in the north to Cap Cerbère in the south and with a water depth ranging 0–90 m (Fig 1). Recreational fishing refers here to all non-commercial fishing activities that are carried out mainly for leisure, where catches are either used for personal consumption, offered to family or friends, or released; the sale of recreational fishing yield being illegal by definition [3]. It encompasses multiple forms of activities performed on- and off-shore, mainly angling, trolling, spearfishing, and shellfish gathering [9, 15, 16].

Surveys consisted of on-site interviews with fishermen in a roving creel survey design [31]. Fishing gear and effort (number and type of lines and hooks, fishing duration, etc.) as well as catch abundance, composition, and size were recorded, including of discarded specimen. The survey instrument was a structured interview featuring a standardized list of questions asked to each participant [30]. Among the multiple approaches that can be used to quantify fishing yields (e.g. scientific campaigns, fisheries logbooks, telephonic surveys) [14, 15, 32], on-site interviews with recreational fishermen have the advantage of maximizing data acquisition in time and space while supporting robust data quality via direct observation of social and ecological descriptors of fisheries (e.g. fishermen abundance, fishing efforts and techniques, catch characteristics). This form of participatory science also promotes positive interactions with fishermen via frequent contact with users (e.g. for increasing awareness and building trust in management strategies). Limitations of such interview-based approaches include the dependency of data quality on user responses to questionnaires. Although the proportion of unreported catch is difficult to evaluate, our interviews indicated that many local fishermen recognized the role of the reserve in preserving marine resources, and it is likely that the majority were honest in their responses. Nevertheless, we assumed the proportion of unreported catch to be relatively constant over time, with no implication on the dynamics of fishing yields as quantified in our study.

Our surveys specifically targeted anglers, who constitute the largest proportion of the local recreational fishing population, fish throughout the year, and are easy to approach for interviews. Fishing gear commonly used by anglers in the study area consist primarily of lures and baited hooks mounted on lines thrown by hand or rod and equipped with weights or floaters [30]. Catch sizes were measured when possible, or otherwise estimated visually or based on fishermen's declarations (e.g. for discarded yields). Catch weights were estimated using length-weight relationships of species from the literature (www.fishbase.org) with locally estimated parameters when available [33]. Fishing efforts (in line.hour) were calculated based on the number of fishing lines and hooks used by each fishermen, and the time spent between when the fishermen declared starting fishing and the interview. Fishing yield was quantified by calculating catch per unit effort (CPUE, number per line per hour) and weight per unit effort (WPUE, gram per line per hour) for all species combined, as well as individually for the three major fish families recorded (Sparidae, Serranidae, Labridae) [34] which represented >85% of catches in number and >65% of the overall weight captured (see S1 Table for a list of the species recorded for each family). CPUE and WPUE are standard metrics of fishing yields per unit of effort, facilitating comparisons of efficiency among different fishing techniques, targets, and regulations [25, 32, 35].

A total of 1,481 interviews were performed between 2005 and 2014, including 493 within the reserve and 988 in surrounding areas (Fig 1). All interviews took place during daytime, and targeted randomly-selected recreational anglers on-shore along the beaches, jetties, and rocky coastline (650 interviews), and off-shore onboard small, typically 4–7 m boats (831 interviews). Interviews were conducted anonymously after informed consent for study participation from each subject.

## Statistical analyses

We used generalized linear models [36] to characterize differences in CPUE and WPUE trajectories between inside and outside the reserve, for fishermen fishing on- and off-shore, for all species catches and each of the 3 major fish families (Sparidae, Serranidae, Labridae). Three-factor models were initially designed to test for differences in yield trajectories (separately for CPUE and WPUE response variables) in time (continuous explanatory variable *Time*: 2005–2014), space (categorical explanatory variable *Reserve*: in vs out), among fishermen groups (categorical explanatory variable *Fishermen*: on- vs off-shore), and their interactions (CPUE ~ *Time* × *Reserve* × *Fishermen*, WPUE ~ *Time* × *Reserve* × *Fishermen*). However, because some of the models did not converge due to over-parametrization [37], simpler two-factor models (*Time* × *Reserve*) were used to characterize CPUE and WPUE trajectories separately for on- and off-shore fishermen. Similar results were found when CPUE and WPUE trajectories were characterized using three-factor models (using data from both fishermen groups) and two-factor models (separately per fishermen group). For consistency, only the latter are reported herein (S2 Table). For clarity and ease of narration, changes in fishing yields as expressed in CPUE and WPUE are described sequentially, from the main effects of the reserve and time alone (Fig 2), to the additional effects of fishermen groups (Fig 3) and fish families (Sparidae, Serranidae, Labridae, Fig 4). A separate set of models was used to compare average CPUE and WPUE values over the entire study period (2005–2014), for all catches and by fish family, between fishing zones and fishermen groups (S3 Table). Preliminary tests of deviance of model residuals indicated a negative-binomial distribution of the data. CPUE and WPUE trajectories were therefore estimated using the glm.nb function with a log link for negative-binomial data from the MASS package [38]. All modeling and graphing were coded in R statistical software (R Core Team). We found similar results when considering fishing effort as number of hooks per hour or number of lines per hour (S1 Fig), only the latter being reported below.

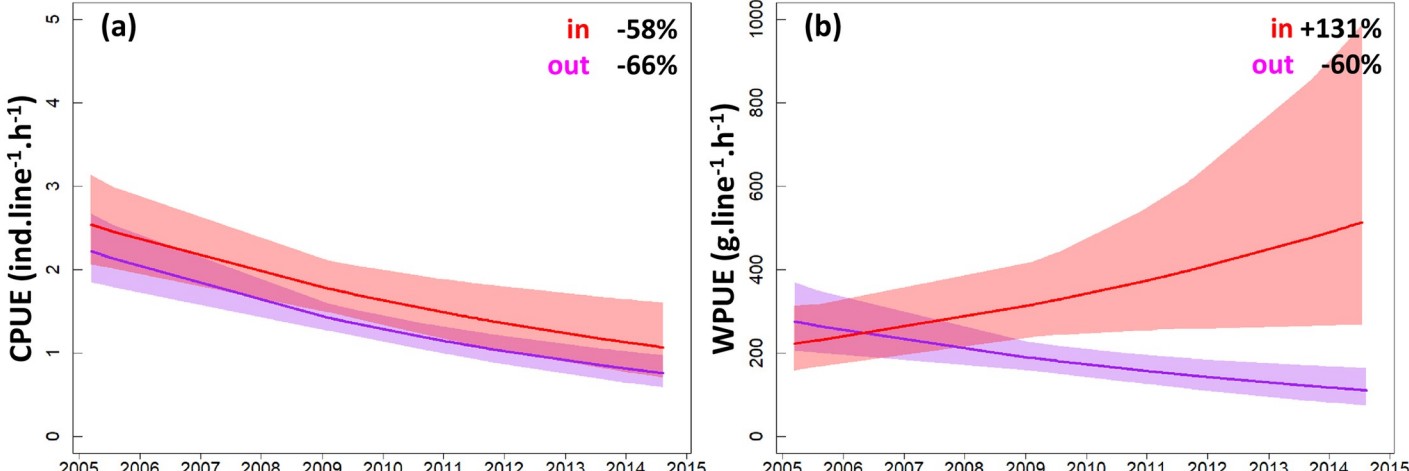

**Fig 2. Trends in catch per unit effort (CPUE, a) and weight per unit effort (WPUE, b) of recreational fishermen fishing inside (in) and outside (out) the Cerbère-Banyuls marine reserve.** Curves represent mean trajectories estimated by generalized linear models and shadings indicate 95% confidence intervals. The percent changes in mean CPUE and WPUE between the beginning and the end of the study period are provided as text on the plots. See S4 Table for mean and confidence interval values. Refer to S5 Table for parameter estimates.

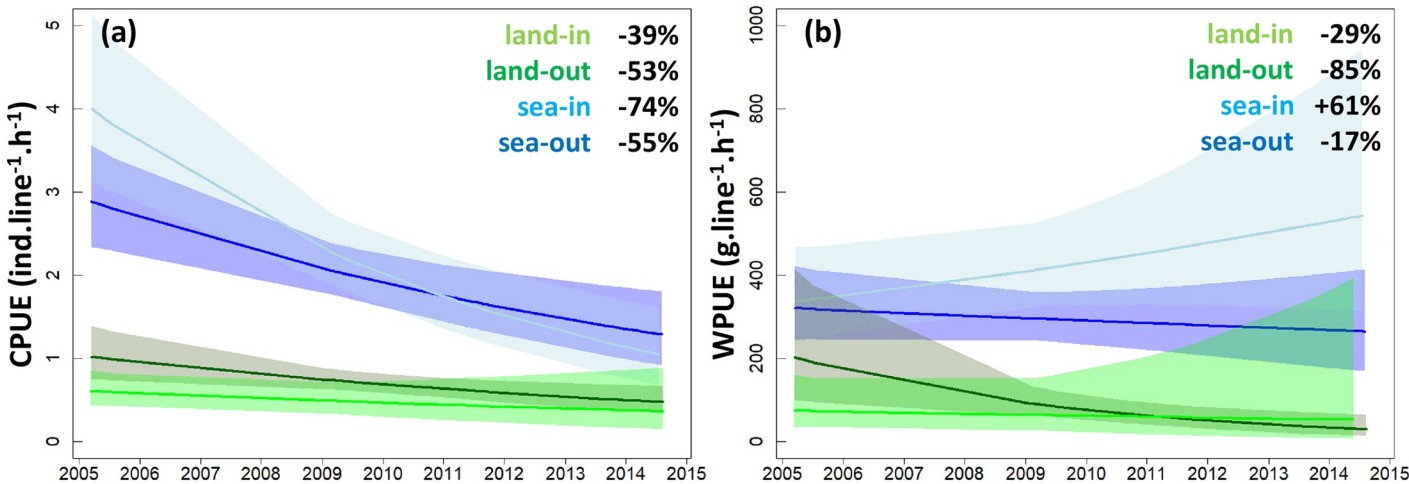

**Fig 3. Trends in catch per unit effort (CPUE, a) and weight per unit effort (WPUE, b) of recreational fishermen fishing from land along the coastline (land) vs at sea from boats (sea), and inside (in) vs outside (out) the Cerbère-Banyuls marine reserve.** Curves represent mean trajectories estimated by generalized linear models and shadings indicate 95% confidence intervals. The percent changes in mean CPUE and WPUE between the beginning and the end of the study period are provided as text on the plots. See S4 Table for mean and confidence interval values. Refer to S5 Table for parameter estimates.

## Results

### All species catches

Over the course of the study, a total of 5,864 individual catches for a fishing effort of 5,028.9 line-hours, and a total fishing yield of 947.0 kg for a fishing effort of 4,970.7 line-hours, were recorded. The average catch per unit effort (CPUE) for the period 2005–2014 was 1.7 ±1.0 SE ind.line$^{-1}$.h$^{-1}$, and the average weight per unit effort (WPUE) was 228.7 ±1.1 SE g.line$^{-1}$.h$^{-1}$. Averaging over all years, fishing yield inside the buffer zone of partial protection in the reserve was 1.4 times higher in terms of catch abundance (CPUE = 2.1 ±1.1 SE vs 1.5 ±1.1 SE ind.line$^{-1}$.h$^{-1}$, *p* = 0.002) and 1.5 times higher in weight, as compared with surrounding areas (WPUE = 288.5 ±1.1 SE vs 198.8 ±1.1 SE g.line$^{-1}$.h$^{-1}$, *p* = 0.018; S3 Table). However, throughout the 2005–2014 survey period, CPUE (Fig 2a) declined both inside (-58%, from 2.5 to 1.1 ind.line$^{-1}$.h$^{-1}$) and outside the reserve (-66%, from 2.2 to 0.8 ind.line$^{-1}$.h$^{-1}$) following a similar pattern to each other, while contrasting WPUE trajectories were observed(Fig 2b), with values increasing in the reserve (+131%, from 222.3 to 514.0 g.line$^{-1}$.h$^{-1}$) and decreasing outside (-60%, from 275.3 to 110.1 g.line$^{-1}$.h$^{-1}$; S4 Table).

The effects of the reserve on CPUE and WPUE trajectories also differed among the two fishermen groups (Fig 3, S4 Table). On-shore, fishing yield was in decline both in the reserve (-39% in CPUE from 0.6 to 0.4 ind.line$^{-1}$.h$^{-1}$, -29% in WPUE from 74.3 to 52.8 g.line$^{-1}$.h$^{-1}$) and in surrounding areas (-53% in CPUE from 1.0 to 0.5 ind.line$^{-1}$.h$^{-1}$, and -85% in WPUE 202.2 to 29.5 g.line$^{-1}$.h$^{-1}$). Off-shore, catch abundance also declined substantially inside (-74% in CPUE from 4.0 to 1.1 ind.line$^{-1}$.h$^{-1}$) and outside the reserve (-55% in CPUE from 2.9 to 1.3 ind.line$^{-1}$.h$^{-1}$), but average yield in weight showed a milder decline outside (-17% in WPUE from 320.7 to 264.7 g.line$^{-1}$.h$^{-1}$) and was increasing in the reserve (+61% in WPUE from 336.7 to 543.4 g.line$^{-1}$.h$^{-1}$). Discriminating catches by fishermen groups alone for the ten-year period revealed on average a 3.6 times lower catch abundance (CPUE = 0.7 ±1.1 SE vs 2.5 ±1.1 SE ind.line$^{-1}$.h$^{-1}$, *p*<0.001) and a 3.9 times lower yield in weight (WPUE = 87.9 ±1.1 SE vs 339.5 ±1.1 SE g.line$^{-1}$.h$^{-1}$, *p*<0.001; S3 Table) from land relative to off-shore.

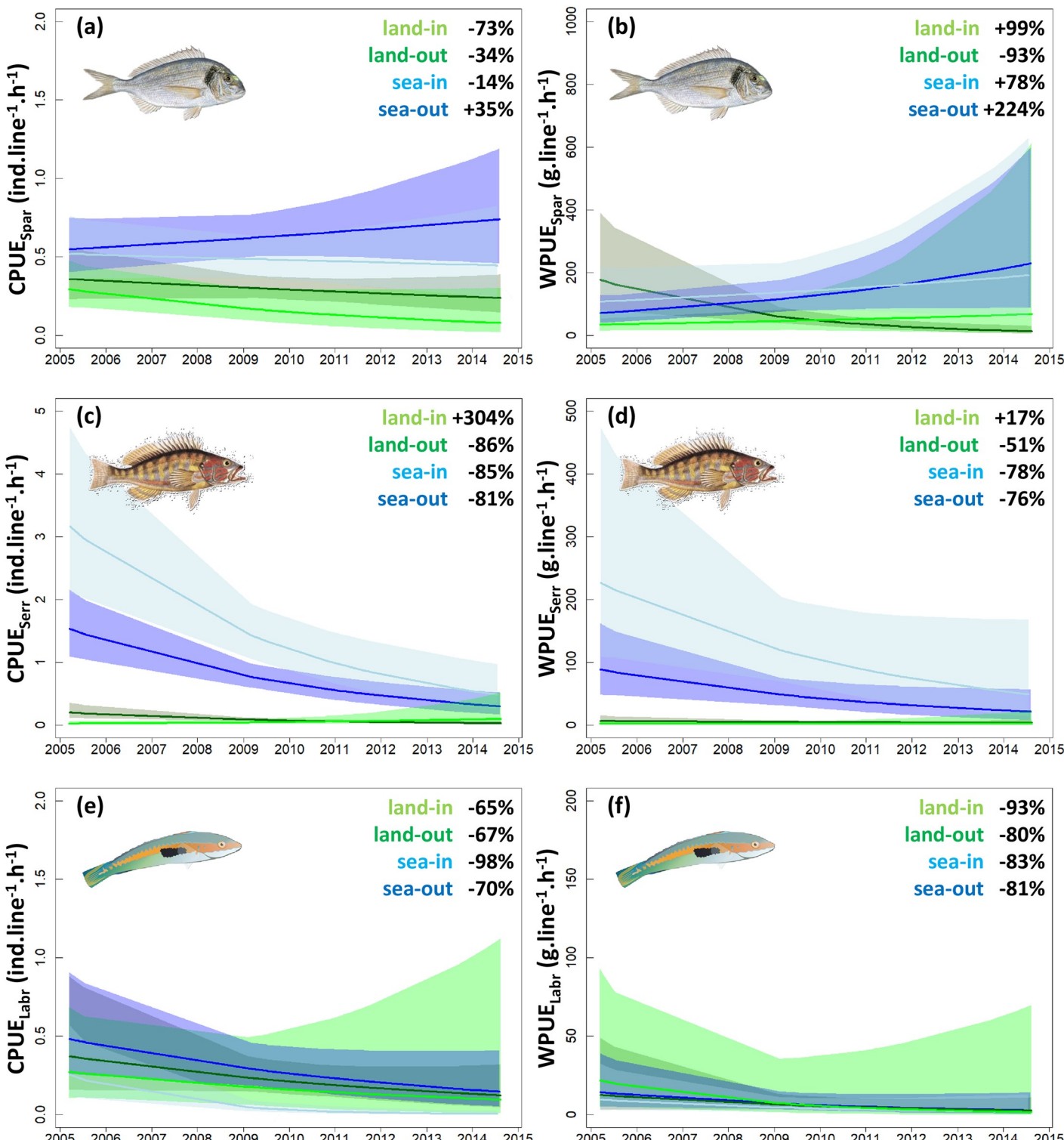

**Fig 4. Trends in catch per unit effort (CPUE) and weight per unit effort (WPUE) of recreational fishermen fishing from land along the coastline (land) vs at sea from boats (sea), and inside (in) vs outside (out) the Cerbère-Banyuls marine reserve for each of the 3 major fish families (Sparidae, Serranidae and Labridae).** Note differences in scale in y-axes. Curves represent mean trajectories estimated by generalized linear models and shadings indicate 95% confidence intervals. The percent changes in mean CPUE and WPUE between the beginning and the end of the study period are provided as text on the plots. See S4 Table for mean and confidence interval values. Refer to S5 Table for parameter estimates.

## Sparidae

Sparidae represented 32% (1,895 ind.) of the total recorded catch, and 47% (440.6 kg) of the overall yield in weight. An average $CPUE_{Spar}$ of 0.4 ±1.1 SE ind.line$^{-1}$.h$^{-1}$ and $WPUE_{Spar}$ of 93.2 ±1.1 SE g.line$^{-1}$.h$^{-1}$ were recorded for 2005–2014 (considering all years). Sparidae catch abundance and weight did not differ significantly between inside and outside the reserve in this period ($CPUE_{Spar}$ = 0.4 ±1.1 SE vs 0.5 ±1.1 SE ind.line$^{-1}$.h$^{-1}$, $p$ = 0.422; $WPUE_{Spar}$ = 100.7 ±1.2 SE vs 89.4 ±1.2 SE g.line$^{-1}$.h$^{-1}$, $p$ = 0.641), but were 2 times higher for fishermen off-shore relative to on-shore ($CPUE_{Spar}$ = 0.6 ±1.1 SE vs 0.3 ±1.1 SE ind.line$^{-1}$.h$^{-1}$, $p<0.001$; $WPUE_{Spar}$ = 119.1 ±1.2 SE vs 60.2 ±1.2 SE g.line$^{-1}$.h$^{-1}$, $p$ = 0.006; S3 Table).

Between 2005 and 2014, average $CPUE_{Spar}$ (Fig 4a) was in decline on-shore both inside (-73%, from 0.3 to 0.1 ind.line$^{-1}$.h$^{-1}$) and outside the reserve (-34%, from 0.4 to 0.2 ind.line$^{-1}$.h$^{-1}$), whereas for off-shore fishermen, a comparatively milder decline was observed in the reserve (-14%, from 0.5 to 0.4 ind.line$^{-1}$.h$^{-1}$) and values were increasing in surrounding areas (+35%, from 0.6 to 0.7 ind.line$^{-1}$.h$^{-1}$; S4 Table). $WPUE_{Spar}$ showed a different pattern over this period (Fig 4b); on-shore values increased in the reserve (+99%, from 34.2 to 68.0 g.line$^{-1}$.h$^{-1}$) but decreased in surrounding non-reserve areas (-93%, from 177.4 to 12.8 g.line$^{-1}$.h$^{-1}$), and off-shore values increased both in the reserve (+78%, from 107.4 to 191.3 g.line$^{-1}$.h$^{-1}$) and in nearby non-reserve waters (+224%, from 70.8 to 229.5 g.line$^{-1}$.h$^{-1}$). Overall, contrasting $WPUE_{Spar}$ trajectories were observed between fishermen performing on- versus off-shore, independently from being located inside or outside of the reserve ($p$ = 0.0013, S2 Table).

## Serranidae

Serranidae represented 42% (2,471 ind.) of the total recorded catch, and 17% (157.4 kg) of the overall yield in weight. An average $CPUE_{Serr}$ of 0.8 ±1.1 SE ind.line$^{-1}$.h$^{-1}$ and $WPUE_{Serr}$ of 55.8 ±1.1 SE g.line$^{-1}$.h$^{-1}$ were recorded for 2005–2014 (considering all years). Serranidae catch abundance and weight were respectively 2.4 and 3 times higher in the reserve relative to surrounding areas in this period ($CPUE_{Serr}$ = 1.4 ±1.2 SE vs 0.6 ±1.1 SE ind.line$^{-1}$.h$^{-1}$, $p<0.001$; $WPUE_{Serr}$ = 99.9 ±1.3 SE vs 33.7 ±1.2 SE g.line$^{-1}$.h$^{-1}$, $p<0.001$), and 17.3 and 23.3 times higher off-shore than on-shore ($CPUE_{Serr}$ = 1.4 ±1.1 SE vs 0.1 ±1.2 SE ind.line$^{-1}$.h$^{-1}$, $p<0.001$; $WPUE_{Serr}$ = 96.6 ±1.2 SE vs 4.1 ±1.2 SE g.line$^{-1}$.h$^{-1}$, $p<0.001$; S3 Table).

Between 2005 and 2014, Serranidae catch abundance and weight (Fig 4c, 4d) declined for off-shore fishermen, both in the reserve (-85% in $CPUE_{Serr}$ from 3.2 to 0.5 ind.line$^{-1}$.h$^{-1}$, -78%$_{Serr}$ in $WPUE_{Serr}$ from 226.5 to 49.0 g.line$^{-1}$.h$^{-1}$) as well as in surrounding areas (-81% in $CPUE_{Serr}$ from 1.5 to 0.3 ind.line$^{-1}$.h$^{-1}$, -76%$_{Serr}$ in $WPUE_{Serr}$ from 88.4 to 21.2 g.line$^{-1}$.h$^{-1}$; S4 Table). Anglers fishing on-shore outside the reserve also experienced declining $CPUE_{Serr}$ (-86%, from 0.2 to 0.0 ind.line$^{-1}$.h$^{-1}$) and $WPUE_{Serr}$ (-51%, from 6.6 to 3.2 g.line$^{-1}$.h$^{-1}$), whereas those fishing on-shore inside the reserve had increasing $CPUE_{Serr}$ (+304%, from 0.0 to 0.1 ind.line$^{-1}$.h$^{-1}$) and $WPUE_{Serr}$ (+17%, from 2.0 to 2.4 g.line$^{-1}$.h$^{-1}$) over time.

## Labridae

Labridae represented 12% (700 ind.) of the total recorded catch, and 3% (25.2 kg) of the overall yield in weight. An average $CPUE_{Labr}$ of 0.2 ±1.1 SE ind.line$^{-1}$.h$^{-1}$ and $WPUE_{Serr}$ of 8.6 ±1.2 SE g.line$^{-1}$.h$^{-1}$ were recorded in 2005–2014 (considering all years). Labridae catch abundance and weight did not differ significantly between inside and outside the reserve in this period ($CPUE_{Labr}$ = 0.2 ±1.3 SE vs 0.3 ±1.2 SE ind.line$^{-1}$.h$^{-1}$, $p$ = 0.058; $WPUE_{Labr}$ = 9.9 ±1.4 SE vs 7.9 ±1.3 SE g.line$^{-1}$.h$^{-1}$, $p$ = 0.616), or among fishermen off-shore as compared with on-shore ($CPUE_{Labr}$ = 0.3 ±1.2 SE vs 0.2 ±1.2 SE ind.line$^{-1}$.h$^{-1}$, $p$ = 0.742; $WPUE_{Labr}$ = 7.9 ±1.3 SE vs 9.4 ±1.4 SE g.line$^{-1}$.h$^{-1}$, $p$ = 0.696; S3 Table).

Labridae catch abundance and weight declined in 2005–2014, both inside and outside of the reserve as well as for both on- and off-shore fishermen (Fig 4e and 4f; S4 Table). Along the shoreline, -65% in $CPUE_{Labr}$ (from 0.3 to 0.1 $ind.line^{-1}.h^{-1}$) and -93% in $WPUE_{Labr}$ (from 21.7 to 1.6 $g.line^{-1}.h^{-1}$) were estimated in the reserve, and -67% in $CPUE_{Labr}$ (from 0.4 to 0.1 $ind.line^{-1}.h^{-1}$) and -80% in $WPUE_{Labr}$ (from 12.5 to 2.5 $g.line^{-1}.h^{-1}$) in surrounding areas. For off-shore fishermen, -98% in $CPUE_{Labr}$ (from 0.3 to 0.0 $ind.line^{-1}.h^{-1}$) and -83% in $WPUE_{Labr}$ (from 9.5 to 1.6 $g.line^{-1}.h^{-1}$) were estimated within the reserve, and -70% in $CPUE_{Labr}$ (from 0.5 to 0.2 $ind.line^{-1}.h^{-1}$) and -81% in $WPUE_{Labr}$ (from 14.2 to 2.7 $g.line^{-1}.h^{-1}$) in surrounding waters.

## Discussion

### Effectiveness of the reserve

Understanding how marine species respond to conservation actions is crucial for successful management of fisheries resources. At the historical site of Cerbère-Banyuls, catch abundances and weights for recreational fishermen were respectively 40% and 50% higher within the buffer zone of partial protection in the reserve than in surrounding areas, indicating significant benefits of the reserve in supporting fishing yield. Fishing restrictions inside the reserve did not protect against the general pattern of decline in catch per unit effort (CPUE) observed in non-reserve areas in 2005–2014, but did support increasing weight per unit effort (WPUE) despite declining values outside the reserve. This indicates changing fishing yields in the reserve through time, with fewer catches overall but an increase in the size of fish that are caught. This finding differs from those reporting increases in catch abundance inside protected areas [12, 26, 35], which might be due to the relatively old age of the Cerbère-Banyuls reserve established in 1974. Indeed, marine reserves promote prolific marine populations, including large predatory species that take longer to re-establish and, through time, are expected to increasingly regulate the abundance of smaller assemblages via trophic cascades [17, 23, 25, 39–41]. For example, the Cerbère-Banyuls marine reserve hosts a recovering population of the large predator dusky grouper (*Epinephelus marginatus*) whose abundance has been increasing from 10 individuals in 1986 to more than 650 in 2020 (the population abundance over our study period being 202 in 2006 and 429 in 2014; www.gemlemerou.org). While the consequences of the loss of large predators for the dynamics of ecosystems is a global concern [42, 43], further investigation is necessary to evaluate the effects of the return of top predators on fisheries resources and marine biota at the Cerbère-Banyuls marine reserve.

### Unequal benefits of the reserve

The benefits of the reserve in promoting fishing yields were limited to off-shore fishermen, whereas on-shore fishermen experienced on average ~4 times lower and declining fishing efficiencies, even within the perimeter of the reserve. While a spatial segregation of larger fish further from the shore could be anticipated, differing trajectories in yield over time indicated reserve benefits were restricted to off-shore fishermen (Fig 3). We did not test for differences in gear characteristics between fishermen (though our estimates of CPUE and WPUE accounted for differences in gear abundance). However, it is unlikely that a same group of users would use significantly different gear inside and outside of the reserve, and that this difference in gear effectiveness would be responsible for the growing yields recorded for off-shore fishermen in the reserve. Alternatively, differences in site accessibility, and therefore fishing pressure, may explain the restriction of reserve benefits to off-shore fishermen [11, 13, 44]. While off-shore fishing is restricted to boat users and segregates fishing effort in a two-dimensional space throughout the reserve, the near-shore is potentially accessible to all fishermen and concentrates fishing pressure on a few accessible sites (mostly along beeches and jetties)

along the mono-dimensional stretch of the coastline (Fig 1). Moreover, near-shore habitats are typically more exposed to other forms of degradation that impact marine biota, including pollution and artificialization of the coastline [2, 6, 45]. Spearfishing, which is often pointed to as a major driver of fish decline in shallow water habitats [12, 25, 46], has been forbidden within the Cerbère-Banyuls marine reserve since 1974, and model simulations indicate that further reducing recreational fishing pressure by 50% could significantly improve stocks of targeted fish species in the area [47, 48]. In 2016, after the period covered by this study, additional restrictions were implemented to regulate recreational fishing pressure within the reserve, anglers being now limited to a maximum of 2 lines with up to 4 hooks if fishing on-shore and 8 hooks off-shore, and the total number of fishermen is now restricted to 1,000 free but mandatory annual permits. While the effects of these new measures on fishing efforts and yields remain to be evaluated, the number of fishing permits could further be reduced in the future as annual user-permit demands have been below the 1,000 threshold in recent years. Nevertheless, there is growing concern that declining fishing yields could jeopardize the popularity of recreational fishing, an emblematic activity in the region, with significant economic consequences for the associated sectors including bait markets, harbors, and tourism.

## Changing catch composition

Spatial differences in fishing yield were found among dominant fish families, suggesting a spatial segregation of fish populations. Average catch abundance and weight in the 10-year period did not differ significantly between inside and outside of the reserve for Labridae and Sparidae, but were 2–3 times higher in the reserve for Serranidae. Similarly, equivalent levels of catch abundance and weight were found on- and off-shore for Labridae, while yields were twice higher off-shore for Sparidae and ~20 times higher for Serranidae. The heterogeneity of benthic habitats was previously found to influence spatial variability in fish assemblage abundance and composition more strongly than protection status at the Cerbère-Banyuls marine reserve [49]. However, contrasting trajectories in fishing yield among fish families over the study period indicated differences in exploitation and/or replenishment of populations and, therefore, changing fish assemblages. For Sparidae, fishing yields in weight increased over time in the reserve as well as in surrounding off-shore areas, while no benefits of the reserve were detected on catch abundances, indicating increasing harvested fish size through time. In contrast, catch abundance and weight declined in all areas for Labridae and Serranidae, except onshore in the reserve where Serranidae catch abundance quadrupled over a decade.

The Sparidae family comprises several large and mobile species that are highly targeted by recreational and commercial fishermen [3, 12, 25, 27, 44, 46], including *Dentex dentex*, *Sparus aurata*, *Lithognathus mormyrus*, *Pagrus pagrus*, and *Diplodus sargus* (S1 Table). As such, the increasing yields recorded for Sparidae indicate the reserve was effective in supporting catches of large individuals from species of high-value to fisheries within the protected area as well as an apparent spillover benefit to adjacent off-shore areas as expected for effective marine reserves [23, 24, 35, 40]. In contrast, the species of Labridae and Serranidae found in the Cerbère-Banyuls marine reserve (S1 Table) are comparatively of low interest to fishermen, which might explain the small differences in yields found between reserve and non-reserve areas [23, 25, 40, 50]. The increasing catch abundance recorded for Serranidae along the shoreline may reflect the capacity of these relatively small, substrate-associated fishes to colonize habitats unoccupied by other species, notably large predatory fish from the family Sparidae [25]. Overall, our decadal-scale evaluation of recreational fishing yields at the Cerbère-Banyuls marine reserve indicates a progressive transfer in catch biomass in space from on-shore to off-shore, as well as in composition from smaller, less-targeted fish to larger species that are of higher value to fisheries.

## Implications for management

Our study shows that surveys of recreational fishing activities can constitute effective alternatives for estimating fishery indicators (e.g. CPUE) compared with using data from commercial fisheries which often have higher uncertainties in declarations on fishing efforts and yields [8, 51, 52]. Our interactions with recreational fishermen also helped create dialogue between scientists, managers, and citizens, building awareness of local management actions to support participation and trust. Our results indicate that, 40 years after its establishment, fishing yields at the Cerbère-Banyuls marine reserve were still changing, implying complex ecological processes that establish on multi-decadal timescales following the creation of a reserve. Over the last decade, changes included shifting catch composition from smaller and less-targeted fish from families Serranidae and Labridae, which with decreasing yields appear in this context as ecological losers among local species, to larger Sparidae fishes that are of high value to fisheries and, with increasing yields, stand as ecological winners. The benefits of the reserve for local fisheries were however mostly restricted to fishermen fishing off-shore who, with increasing yields, stand as social winners of the current management plan, whereas fishermen on-shore, the social losers, suffered ~4 times lower and declining fishing yields. In addition, large confidence intervals surrounding estimates of fishing yields over recent years indicate that recorded increases in yields for off-shore fishermen may not be equally shared among users (Figs 2–4).

With mean CPUE ranging 0.4–4.0 ind.line$^{-1}$.h$^{-1}$ and mean WPUE ranging 30.5–543.4 g. line$^{-1}$.h$^{-1}$ (S4 Table), recreational fishing yields at the Cerbère-Banyuls marine reserve are within the range of those reported along the north-western Mediterranean coast [9]. Similarly, the decline in shallow-water fishing yield observed locally reflects the broader pattern of declining near-shore yields at the scale of the entire Mediterranean [1–3]. As such, local management outcomes at Cerbère-Banyuls can help define regional plans, though identifying how local success at the small scale of the reserve (650 ha) can be expanded to the broader scale of the marine park (4,010 km$^2$) or that of the entire Gulf of Lion (Fig 1) remains a challenge. The possibility of multiplying the number of natural reserves like Cerbère-Banyuls to amplify marine protected area benefits and counter declining fishing yields and coastal degradation in the region is presently under discussion [53, 54]. Nevertheless, while the increasing yields for Sparidae attest for the positive effects of the reserve on stocks of targeted fish species, the decline of near-shore yields poses several challenges, including that of preserving the socio-economic benefits of fishery activities (both recreational and commercial) and maintaining access to resources for different user groups (both on- and off-shore), for which group-specific regulations could be enforced [3, 10, 11, 39]. Our findings indicate that current management plans do not benefit on-shore fishing, undermining equity in this emblematic activity that is historically accessible to all and particularly popular among vacationers, low-income, and retired people. Given increasing pressure on common-pool natural resources and growing socio-economic inequalities, this emerging issue needs to be prioritized in sustainable management actions [19, 22, 55].

## Supporting information

**S1 Table. List of the species recorded for each of the three major fish families caught by recreational fishermen around the Cerbère-Banyuls natural marine reserve.**
(PDF)

**S2 Table. Deviance table of generalized linear models characterizing changes in catch per unit effort and weight per unit effort of on-shore and off-shore recreational fishing, inside versus outside the Cerbère-Banyuls marine reserve.**
(PDF)

**S3 Table. Deviance table of generalized linear models comparing average catch per unit effort and weight per unit effort of on-shore and off-shore recreational fishing, inside versus outside the Cerbère-Banyuls marine reserve over the entire study period.**
(PDF)

**S4 Table. Yields in catch per unit effort and weight per unit effort estimated for on-shore and off-shore recreational fishing, inside and outside of the Cerbère-Banyuls marine reserve at the beginning and end of the survey.**
(PDF)

**S5 Table. Parameter estimates of generalized linear models characterizing trends in catch per unit effort and weight per unit effort of on-shore and off-shore recreational fishing, inside versus outside the Cerbère-Banyuls marine reserve.**
(PDF)

**S1 Fig. Trends in catch per unit effort and weight per unit effort of recreational fishing inside versus outside the Cerbère-Banyuls marine reserve with unit effort expressed in line.hours and in hook.hours.**
(PDF)

## Acknowledgments

We would like to thank all the recreational fishermen who accepted to participate in our interviews. We also thank F. Huguet, C. Pou, C. Marjorie, F. Mirbeau, J. Jean-Baptiste, R. Neveu, A. Gudefin, and A. Missa for participating in some field trips, and J. Pastor for helping with boating logistics. We thank the managers of the Cerbère-Banyuls Natural Marine Reserve, notably M.-L. Licari, J.-F. Laffon, and F. Cadene, as well as of the Natural Marine Park of the Gulf of Lion, G. Le Corre and H. Magnin, for supporting the realization of our studies. We also thank the municipalities of Argelès-sur-Mer and Banyuls-sur-Mer for facilitating our surveys by providing us port space. We are grateful to J. Ballard for improvements on the manuscript.

## Author Contributions

**Conceptualization:** Mohsen Kayal, Marine Cigala, Manon Mercader, Virginie Hartmann, Philippe Lenfant, Dominique Pelletier, Marion Verdoit-Jarraya.

**Data curation:** Mohsen Kayal, Marine Cigala, Nelly Soulat, Audrey Lebras, Virginie Hartmann.

**Formal analysis:** Mohsen Kayal.

**Funding acquisition:** Philippe Lenfant, Dominique Pelletier, Marion Verdoit-Jarraya.

**Investigation:** Mohsen Kayal, Marine Cigala, Nelly Soulat, Manon Mercader, Audrey Lebras, Pauline Ivanoff, Léa Sébési, Aurélie Lassus-Debat, Virginie Hartmann, Mélissa Bradtke, Julien Dubreuil, Manon Joguet, Solène Le Mellionnec, Marion Brichet, Jean-Louis Binche, Jérôme Payrot, Gilles Saragoni, Romain Crec'hriou, Marion Verdoit-Jarraya.

**Methodology:** Mohsen Kayal, Nelly Soulat, Manon Mercader, Philippe Lenfant, Dominique Pelletier, Marion Verdoit-Jarraya.

**Project administration:** Dominique Pelletier, Marion Verdoit-Jarraya.

**Resources:** Nelly Soulat, Virginie Hartmann, Philippe Lenfant, Coraline Jabouin, Gilles Saragoni, Romain Crec'hriou, Marion Verdoit-Jarraya.

**Software:** Eléonore Cambra.

**Supervision:** Philippe Lenfant, Coraline Jabouin, Dominique Pelletier, Marion Verdoit-Jarraya.

**Validation:** Marion Verdoit-Jarraya.

**Visualization:** Mohsen Kayal, Marine Cigala, Eléonore Cambra.

**Writing – original draft:** Mohsen Kayal.

**Writing – review & editing:** Mohsen Kayal, Marion Verdoit-Jarraya.

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
