## [Decision Letter · Decision Letter 0]

30 Sep 2020

PONE-D-20-23582

Marine reserve benefits and recreational fishing yields: the winners and the losers

PLOS ONE

Dear Dr. Kayal,

Thank you for submitting your manuscript to PLOS ONE. After careful consideration, we feel that it has merit but does not fully meet PLOS ONE’s publication criteria as it currently stands. Therefore, we invite you to submit a revised version of the manuscript that addresses the points raised during the review process.

In your revision please address carefully the comments/suggestions made by reviewer #1 regarding the background information on the modelling as well as the presentation and interpretation of the results. One other important aspect that need to be address for your manuscript to be considered further for publication in PLOS ONE is the availability of the underlying data, so please explain more clearly why ALL the data can not be made fully available or if possible they can be access via a special request.

We look forward to receiving your revised manuscript.

Kind regards,

Andrea Belgrano, Ph.D.

Academic Editor

PLOS ONE

Journal Requirements:

'This study was funded by multiple sources including the French Agency for Biodiversity (https://ofb.gouv.fr) and Region Occitanie/Pyrénées-Méditerranée (www.laregion.fr), Département des Pyrénées-Orientales (www.ledepartement66.fr), French Ministry of Ecology, Energy, Sustainable Development and Regional Planning (www.ecologique-solidaire.gouv.fr), European Fisheries Fund (https://ec.europa.eu), Pays Pyrénées Méditerranée (www.payspyreneesmediterranee.org), as well as the Regional Direction for the Environment, Planning and Housing Occitanie (www.occitanie.developpement-durable.gouv.fr). Data were collected in part during the PAMPA project on Indicators of MPA Performance funded by the French Ministry of Ecology and the French MPA Agency. The funders had no role in study design, data collection and analysis, decision to publish, or preparation of the manuscript.'

We note that one or more of the authors are employed by a commercial company: SEANEO.

4. We note that Figure 1 in your submission contains a map image which may be copyrighted.

We require you to either (a) present written permission from the copyright holder to publish these figure specifically under the CC BY 4.0 license, or (b) remove the figure from your submission:

b. If you are unable to obtain permission from the original copyright holder to publish these figure under the CC BY 4.0 license or if the copyright holder’s requirements are incompatible with the CC BY 4.0 license, please either i) remove the figure or ii) supply a replacement figure that complies with the CC BY 4.0 license. Please check copyright information on all replacement figures and update the figure caption with source information. If applicable, please specify in the figure caption text when a figure is similar but not identical to the original image and is therefore for illustrative purposes only.

Reviewers' comments:

Reviewer's Responses to Questions

**Comments to the Author**

1. Is the manuscript technically sound, and do the data support the conclusions?

Reviewer #1: Partly

2. Has the statistical analysis been performed appropriately and rigorously? 

Reviewer #1: I Don't Know

3. Have the authors made all data underlying the findings in their manuscript fully available?

Reviewer #1: No

4. Is the manuscript presented in an intelligible fashion and written in standard English?

Reviewer #1: Yes

5. Review Comments to the Author

Reviewer #1: This paper uses surveys of recreational anglers over a period of ten years to derive several indicators of fishing yield (Catch per unit effort (CPUE) and weight per unit effort (WPUE), in various zones in and around a marine reserve. The objective is to determine whether the reserve is resulting in improved fish stock status and recreational fishing yields, and to whom these benefits are distributed. They found that the average CPUE and WPUE per year were higher inside the reserve, but that over time, CPUE has declined both inside, and outside the reserve. However, they did find that WPUE is increasing inside the reserve, but declining outside. The increase in WPUE is only being experienced by offshore fishers in boats however, while the onshore anglers continue to experience a decline in WPUE, regardless of whether they are fishing inside or outside the reserve. This data has the potential to be a valuable source of evidence for decision making and the introduction and conclusion are well written, however the paper is missing some important details about the modelling methods and survey data, which make it difficult to assess the validity of the results obtained.

Main comments

My main concern is around the level of detail provided in the methods, particularly around the modelling. The methods state that the models are used to compare CPUE/WPUE in different zones and for species, however a lack of detail in the methods section means it is unclear how this was achieved, making it difficult to determine how robust the reported results are. More specifically:

The objective of the modelling is unclear: while the methods state the objective is to identify differences between different catch conditions, the results (eg Fig 1) show the models have been used to model trajectories of CPUE/WPUE for different groups, which were then compared (somehow) to ascertain their difference.

More information is needed on how many models were fitted (is there a separate model for each zone? taxonomic grouping?) and with what response and explanatory variables. For example:

- Is the response variable the CPUE/WPUE, subset by onshore/offshore etc, or is it the difference in CPUE/WPUE between onshore and offshore fishers?

- What are the explanatory variables considered, and retained in the final selected model? Are these the variables listed as ‘factors’ in Table S2?

Some details are provided in Table S2, but the caption, and particularly column names, are not informative. These issues could be relatively easily addressed by:

1. Explicitly stating the model’s purpose and response variables in the methods.

2. Explaining the model selection process undertaken in the methods

3. Including a tables of candidate, and selected, models in the supporting information to replace existing Table S2, along with specifically labelled explanatory and response variables (final models could even go in a table in the results depending on how many there are). (For examples of how to address points 2 and 3, you could refer to the main text and supporting information in Curnick DJ, Collen B, Koldewey HJ, Jones KE, Kemp KM, Ferretti F. 2020. Interactions between a large marine protected area, pelagic tuna and associated fisheries. Frontiers in Marine Science 7:318. )

4. Include some references to the literature around the methods used (general GLM literature as well as their specific use in fisheries for estimating indicators)

Estimation of CPUE and comparison of groups

Throughout the results, the authors compare average CPUE and WPUE between zones (onshore/offshore, in reserve/outside reserve), and provide a p value as evidence of statistical significance (eg line 239, 262 – 265 etc). It is not clear exactly how this p value was obtained, and they do not match p values reported in Table S4.

Please include a few sentences in the methods re. how the authors went from the models, to estimating the trajectories, and what test they used to compare average values to obtain the p values reported in the results.

Survey data

WPUE (Fig. 2 – 4 panel b) exhibits a marked expansion in confidence intervals post 2009. This pattern - combined with this result being somewhat unexpected (lines 351 – 352), and also informing some of the key arguments in the discussion - warrants some further attention in the discussion, along with some more information about the raw survey data (cleaning processes, figures showing the distribution – e.g. boxplots in the supporting information or as points in Figures 2 – 5)

Minor comments

Reporting of indicator values

The paper reports several different indicators across different zones (eg inshore/offshore, in reserve/outside reserve): Average catch abundance per year during the survey period, Average catch weight per year during the survey period (eg line 239 CPUE = 2.1 + 1.1 SE ind.line-1.h-1), CPUE trajectory over time, WPUE trajectory over time (eg +131%, from 222.3 to 514.0 g.line-1.h-1)

Because the authors are comparing the metrics over both space and time, it needs to be explicit which numbers refer to which indicator and type of comparison. Currently several different numbers are reported for CPUE but it is difficult to ascertain what the difference between these metrics are (some numbers refer to spatial averages, and others refer to trends over time). For example, rather than “fishing yield … was 1.4 times higher in terms of catch abundance (CPUE=2.1 ±1.1 SE vs 1.5 ±1.1 SE ind.line-1.h-1, p=0.002)” (lines 238 – 239), it would be clearer as something like “During the survey period, average catch abundance per year … was 1.4 times higher inside the reserve than outside (Ave. CPUE/year = 2.1 ±1.1 SE …)”.

Likewise, when comparing trajectories, explain that the decline is over time (rather than through space). This applies throughout the results section.

Lines 45 – 46 & 367 – 391: Discussion of unequal benefits. Could differences in gear (size/weight/type, rather than number of lines/hooks), or distribution of larger fish be another explanation for the differences in WPUE achieved by onshore/offshore fishers? If this is the case, further regulation of the reserve would be unlikely to achieve equitable distribution of benefits, so it may be important to cover this off before recommending regulation.

Lines 35 & 242: Begin by saying CPUE showed a ‘similar pattern’ – I think the objective was to explain that the CPUE trajectory is similar in both areas (in and out reserve), but comes across as saying the CPUE is similar to the results reported in the previous sentence. This gets very confusing as the previous sentence was reporting a positive effect. Suggest altering to something along the lines of ‘CPUE showed a declining trend over time, both inside and outside the reserve (reported figures here)’.

Lines 77 – 81: This paragraph provides important rationale for the research, but the logic could be articulated more explicitly – why do uncertainties about benefits pose regulatory challenges? How does identifying social and ecological winners address this challenge? How does it contribute to adaptive management? In addition, the discussion doesn’t address adaptive management specifically.

Lines 131 – 134: Description of the reserve’s value includes some unnecessary superlatives (e.g. exemplary, exceptional) which affect the objective tone used elsewhere.

Lines 189 – 191: The authors have made several assumptions about potential flaws in collecting survey data (1 – that respondees were honest, and 2 – that proportion of unreported catch remains consistent over time). These may be accepted assumptions for this method, but could be better supported with evidence/references and warrant a mention in the discussion.

Line 221: Variable ‘Time’ needs a unit – e.g. year, month, day?

Line 343 – 346 & 32: “At the historical site of Cerbère-Banyuls, catch abundances and weights for recreational fishermen were respectively 40% and 50% higher within the buffer zone of partial protection in the reserve than in surrounding areas, indicating significant benefits of the reserve in supporting fishing yield”. How is ‘significant benefit ’defined? Authors should rather discuss this in the context of the subsequent metrics, which indicate that while the overall CPUE may be higher inside the reserve, it is still declining steadily over time, so the benefits are not unequivocal. E.g., discuss the reserve benefits, with caveats, after reporting the other metrics as well.

Line 429 – 432: states that “ Our study shows that surveys of recreational fishing activities can constitute robust alternatives for estimating fishery indicators (e.g. CPUE) compared with using data from commercial fisheries …” , This statement should be rephrased to more accurately represent what was achieved. As no comparison is made with either commercial data or fisheries independent data, it is not possible to assess robustness, but the paper does demonstrate the use of a worthwhile alternate approach.

Lines 441 – 442: The terms social and ecological winners appear somewhat out of the blue here (although social and ecological protagonists are mentioned in the introduction). Defining these concepts and their implications in the introduction would strengthen this conclusion when it appears.

6. PLOS authors have the option to publish the peer review history of their article (what does this mean?). If published, this will include your full peer review and any attached files.

Reviewer #1: No

---

## [Author Response · Author response to Decision Letter 0]

12 Nov 2020

Editor’s comments:

In your revision please address carefully the comments/suggestions made by reviewer #1 regarding the background information on the modelling as well as the presentation and interpretation of the results. 

Additional information is now provided on the modelling methods, results, and interpretation (see response to reviewer comments below).

One other important aspect that need to be address for your manuscript to be considered further for publication in PLOS ONE is the availability of the underlying data, so please explain more clearly why ALL the data can not be made fully available or if possible they can be access via a special request.

We have now posted the raw data used in the study on the Zenodo digital repository (http://doi.org/10.5281/zenodo.4149014).

We revised the manuscript following PLOS ONE article styles.

2. We note that you have indicated that data from this study are available upon request. PLOS only allows data to be available upon request if there are legal or ethical restrictions on sharing data publicly. If there are ethical or legal restrictions on sharing a de-identified data set, please explain them in detail (e.g., data contain potentially sensitive information, data are owned by a third-party organization, etc.) and who has imposed them (e.g., an ethics committee). Please also provide contact information for a data access committee, ethics committee, or other institutional body to which data requests may be sent.

We have now posted the raw data used in the study on the Zenodo digital repository (http://doi.org/10.5281/zenodo.4149014).

'This study was funded by multiple sources including the French Agency for Biodiversity (https://ofb.gouv.fr) and Region Occitanie/Pyrénées-Méditerranée (www.laregion.fr), Département des Pyrénées-Orientales (www.ledepartement66.fr), French Ministry of Ecology, Energy, Sustainable Development and Regional Planning (www.ecologique-solidaire.gouv.fr), European Fisheries Fund (https://ec.europa.eu), Pays Pyrénées Méditerranée (www.payspyreneesmediterranee.org), as well as the Regional Direction for the Environment, Planning and Housing Occitanie (www.occitanie.developpement-durable.gouv.fr). Data were collected in part during the PAMPA project on Indicators of MPA Performance funded by the French Ministry of Ecology and the French MPA Agency. The funders had no role in study design, data collection and analysis, decision to publish, or preparation of the manuscript.'

We note that one or more of the authors are employed by a commercial company: SEANEO.

Thank you for these considerations about potential competing interests. In fact, contributing author N.S. participated in several of the survey campaigns of this study while under affiliations 1 & 2 (host laboratory at the University of Perpignan). We therefore amended her affiliations on the manuscript, indicating her previous affiliations with the host laboratory, and mentioning the private company SEANEO as her present address. Our competing interest statement therefore remains unchanged.

We also updated the affiliations for the three coauthors A.L.-D., J.D., and M.J. who were in similar cases (contributions to study under affiliations 1 & 2, with new current addresses).

4. We note that Figure 1 in your submission contains a map image which may be copyrighted.

We require you to either (a) present written permission from the copyright holder to publish these figure specifically under the CC BY 4.0 license, or (b) remove the figure from your submission:

b. If you are unable to obtain permission from the original copyright holder to publish these figure under the CC BY 4.0 license or if the copyright holder’s requirements are incompatible with the CC BY 4.0 license, please either i) remove the figure or ii) supply a replacement figure that complies with the CC BY 4.0 license. Please check copyright information on all replacement figures and update the figure caption with source information. If applicable, please specify in the figure caption text when a figure is similar but not identical to the original image and is therefore for illustrative purposes only.

The maps in Fig 1 were produced using the open source program QGIS. This information is now provided in the figure caption (l. 103). 

(1) Thank you for including your ethics statement on the online submission form: 

This study involved interviews with recreational fishermen. The interviews were

conducted anonymously after informed consent for study participation from each

subject. The survey methodology and material was approved by the University of

Perpignan.

 To help ensure that the wording of your manuscript is suitable for publication, would you please also add this statement at the beginning of the Methods section of your manuscript file.

This statement has been added at the beginning of the Methods section (l. 120-123). 

(2) Thank you for providing additional information regarding the authors' affiliations. Could you please clarify whether author NS was affiliated with SEANO at the time of study?

Contributing author N.S. participated to this study while under affiliations 1 & 2 only (host laboratory at the University of Perpignan), with no affiliation with the commercial company SEANEO at the time of the study.

Reviewers' comments:

Reviewer #1: This paper uses surveys of recreational anglers over a period of ten years to derive several indicators of fishing yield (Catch per unit effort (CPUE) and weight per unit effort (WPUE), in various zones in and around a marine reserve. The objective is to determine whether the reserve is resulting in improved fish stock status and recreational fishing yields, and to whom these benefits are distributed. They found that the average CPUE and WPUE per year were higher inside the reserve, but that over time, CPUE has declined both inside, and outside the reserve. However, they did find that WPUE is increasing inside the reserve, but declining outside. The increase in WPUE is only being experienced by offshore fishers in boats however, while the onshore anglers continue to experience a decline in WPUE, regardless of whether they are fishing inside or outside the reserve. This data has the potential to be a valuable source of evidence for decision making and the introduction and conclusion are well written, however the paper is missing some important details about the modelling methods and survey data, which make it difficult to assess the validity of the results obtained.

Thank you for your in depth evaluation of our manuscript, and pointing out these issues. We have now revised the descriptions of our statistical analyses, clarified our overall study approach, and amended our interpretation of the results in the light of the comments (as detailed below).

Main comments

My main concern is around the level of detail provided in the methods, particularly around the modelling. The methods state that the models are used to compare CPUE/WPUE in different zones and for species, however a lack of detail in the methods section means it is unclear how this was achieved, making it difficult to determine how robust the reported results are. More specifically:

The objective of the modelling is unclear: while the methods state the objective is to identify differences between different catch conditions, the results (eg Fig 1) show the models have been used to model trajectories of CPUE/WPUE for different groups, which were then compared (somehow) to ascertain their difference.

More information is needed on how many models were fitted (is there a separate model for each zone? taxonomic grouping?) and with what response and explanatory variables. For example:

- Is the response variable the CPUE/WPUE, subset by onshore/offshore etc, or is it the difference in CPUE/WPUE between onshore and offshore fishers?

- What are the explanatory variables considered, and retained in the final selected model? Are these the variables listed as ‘factors’ in Table S2?

Some details are provided in Table S2, but the caption, and particularly column names, are not informative. These issues could be relatively easily addressed by:

1. Explicitly stating the model’s purpose and response variables in the methods.

A set of GLMs were indeed used to estimate trajectories of the response variables CPUE and WPUE inside versus outside of the reserve, and to identify temporal changes in yield for fishermen groups (on- and off-shore) and fish-families (Sparidae, Serranidae, Labridae). This general purpose is stated l. 212-215. 

We initially implemented three-factor models (CPUE ~ Time × Reserve × Fishermen, and WPUE ~ Time × Reserve × Fishermen) to test for differences in reserve effects on trajectories among fishermen groups. However, lack of convergence indicated over-parametrization of some of the models, which we therefore split into simpler two-factor models (Time × Reserve) applied separately to on- and off-shore fishermen data). This is now specified l. 215-222.

For consistency, we report two-factor models (Time × Reserve, now better detailed in S2 Table) characterizing CPUE and WPUE trajectories for all species and fishermen (Fig. 2, model set 1 in S2 Table), and separately per fishermen group (Fig. 3, model sets 2-3 in S2 Table) and per fish family (Fig. 4, model sets 4-9 in S2 Table). We now also specify in the text that similar results were found when trajectories were estimated using two- and three-factor models, only the latter being reported in the manuscript (l. 222-225).

2. Explaining the model selection process undertaken in the methods

3. Including a tables of candidate, and selected, models in the supporting information to replace existing Table S2, along with specifically labelled explanatory and response variables (final models could even go in a table in the results depending on how many there are). (For examples of how to address points 2 and 3, you could refer to the main text and supporting information in Curnick DJ, Collen B, Koldewey HJ, Jones KE, Kemp KM, Ferretti F. 2020. Interactions between a large marine protected area, pelagic tuna and associated fisheries. Frontiers in Marine Science 7:318. )

As described above, we have now amended the manuscript text and S2 Table to clarify our statistical approach and design of the GLMs characterizing fishing yield trajectories. This includes a clear identification of explanatory and response variables (l. 215-222).

We did not perform additional model selection as used in GAMM modeling, though we now cite the suggested study by Curnick et al. in relation to the effects of the reserve for higher trophic level species (citation #41, manuscript l. 363).

4. Include some references to the literature around the methods used (general GLM literature as well as their specific use in fisheries for estimating indicators)

In addition to the already cited work by Ripley et al. 2019 on the MASS package from which the GLM function was used (citation #38 l. 233), we have now added reference to the book by Dunn & Smyth (2018) that provides numerous examples of GLM applications (citation #36 l. 212). We also added reference to work by Harrison et al. (2018) discussing the mentioned over-parametrization issue in GLMs (citation #37, l. 221).

Estimation of CPUE and comparison of groups

Throughout the results, the authors compare average CPUE and WPUE between zones (onshore/offshore, in reserve/outside reserve), and provide a p value as evidence of statistical significance (eg line 239, 262 – 265 etc). It is not clear exactly how this p value was obtained, and they do not match p values reported in Table S4.

Please include a few sentences in the methods re. how the authors went from the models, to estimating the trajectories, and what test they used to compare average values to obtain the p values reported in the results.

Average CPUE and WPUE over the entire study period (pooled over 2005-2014) were compared using a separate set of GLMs (different from those used to estimate trajectories). We previously did not detail this to avoid overloading of the manuscript, but we now include information on these tests in the Methods section l. 228-230 as well as in our new S3 Table.

Survey data

WPUE (Fig. 2 – 4 panel b) exhibits a marked expansion in confidence intervals post 2009. This pattern - combined with this result being somewhat unexpected (lines 351 – 352), and also informing some of the key arguments in the discussion - warrants some further attention in the discussion, along with some more information about the raw survey data (cleaning processes, figures showing the distribution – e.g. boxplots in the supporting information or as points in Figures 2 – 5)

Raw data on fishing yields are illustrated in S1 Fig. Yield data were characterized by high dispersion, which was taken into account in our analyses with the use of a GLM function for negative-binomial distribution (mentioned l. 231-233). 

We preferred presenting average curves ±confidence-intervals in our figures in the manuscript, rather than average curves with raw data, given the difficulty to read the latter graphs due to broad axis value ranges (as illustrated in S1 Fig).

However, the remark of the reviewer on the increase in confidence interval in time is indeed interesting, as it indicates further dispersion of the data with years, or in other words, increasing disparity in yield among individual observations. This indicates the identified average increases in yields may not be equally distributed among off-shore fishermen, and is now mentioned in Discussion l. 452-454.

Minor comments

Reporting of indicator values

The paper reports several different indicators across different zones (eg inshore/offshore, in reserve/outside reserve): Average catch abundance per year during the survey period, Average catch weight per year during the survey period (eg line 239 CPUE = 2.1 + 1.1 SE ind.line-1.h-1), CPUE trajectory over time, WPUE trajectory over time (eg +131%, from 222.3 to 514.0 g.line-1.h-1)

Because the authors are comparing the metrics over both space and time, it needs to be explicit which numbers refer to which indicator and type of comparison. Currently several different numbers are reported for CPUE but it is difficult to ascertain what the difference between these metrics are (some numbers refer to spatial averages, and others refer to trends over time). For example, rather than “fishing yield … was 1.4 times higher in terms of catch abundance (CPUE=2.1 ±1.1 SE vs 1.5 ±1.1 SE ind.line-1.h-1, p=0.002)” (lines 238 – 239), it would be clearer as something like “During the survey period, average catch abundance per year … was 1.4 times higher inside the reserve than outside (Ave. CPUE/year = 2.1 ±1.1 SE …)”.

Likewise, when comparing trajectories, explain that the decline is over time (rather than through space). This applies throughout the results section.

Thank you. We added further clarifying terms throughout the Results section to account for this comment (l. 244, 248, 285, 315, 333).

Lines 45 – 46 & 367 – 391: Discussion of unequal benefits. Could differences in gear (size/weight/type, rather than number of lines/hooks), or distribution of larger fish be another explanation for the differences in WPUE achieved by onshore/offshore fishers? If this is the case, further regulation of the reserve would be unlikely to achieve equitable distribution of benefits, so it may be important to cover this off before recommending regulation.

While a segregation of larger fish further from the shore could be anticipated (that is, larger average WPUE off-shore), our estimations of fishing yield trajectories over time indicated the reserve benefits (increasing yields) were restricted to off-shore fishermen (now stated l. 375-377).

We did not test for differences in gear characteristics between fishermen (our estimates of CPUE and WPUE already accounting for differences in gear abundance). However, it is unlikely that a same group of users would use significantly different gear when fishing inside versus outside of the reserve, and that this difference in gear effectiveness would be responsible for the growing yields recorded for off-shore fishermen inside the reserve (now stated l. 377-382). 

Lines 35 & 242: Begin by saying CPUE showed a ‘similar pattern’ – I think the objective was to explain that the CPUE trajectory is similar in both areas (in and out reserve), but comes across as saying the CPUE is similar to the results reported in the previous sentence. This gets very confusing as the previous sentence was reporting a positive effect. Suggest altering to something along the lines of ‘CPUE showed a declining trend over time, both inside and outside the reserve (reported figures here)’.

This has been corrected (l. 36 and 247-252).

Lines 77 – 81: This paragraph provides important rationale for the research, but the logic could be articulated more explicitly – why do uncertainties about benefits pose regulatory challenges? 

The sentence now specifies in terms of implementing appropriate measures for resource durability and equitable access (l. 73-74).

How does identifying social and ecological winners address this challenge? How does it contribute to adaptive management? In addition, the discussion doesn’t address adaptive management specifically.

We have now changed this sentence to “… can refine regulatory strategies and help define win-win sustainable management for people and ecosystems” (l. 74-76).

Lines 131 – 134: Description of the reserve’s value includes some unnecessary superlatives (e.g. exemplary, exceptional) which affect the objective tone used elsewhere.

These superlatives have been removed.

Lines 189 – 191: The authors have made several assumptions about potential flaws in collecting survey data (1 – that respondees were honest, and 2 – that proportion of unreported catch remains consistent over time). These may be accepted assumptions for this method, but could be better supported with evidence/references and warrant a mention in the discussion.

There are no studies evaluating unreported catch of recreational fishermen in our study area. Similarly, there were no reasons to postulate that unreported catch would change over time over our study period. It is our preference to report these assumptions at once in the Methods section, and focus the Discussion on the findings of the study.

Line 221: Variable ‘Time’ needs a unit – e.g. year, month, day?

We now provide a range in years in the sentence: 2005-2014.

Line 343 – 346 & 32: “At the historical site of Cerbère-Banyuls, catch abundances and weights for recreational fishermen were respectively 40% and 50% higher within the buffer zone of partial protection in the reserve than in surrounding areas, indicating significant benefits of the reserve in supporting fishing yield”. How is ‘significant benefit ’defined? Authors should rather discuss this in the context of the subsequent metrics, which indicate that while the overall CPUE may be higher inside the reserve, it is still declining steadily over time, so the benefits are not unequivocal. E.g., discuss the reserve benefits, with caveats, after reporting the other metrics as well.

Given the multiple aspects of our study on reserve benefits for fish and fishermen groups, we opted for a hierarchical narrative in the discussion: starting from the simple effect of the reserve alone averaged over the 10 year period (l. 351-354), and unfolding with the sequential mention of differences in trajectories among fishing areas (l. 354-357), fishermen groups (l. 373-375) and fish families (l. 405-410). 

Overall, it is hard to conclude if the reserve was not beneficial because catch numbers were declining, or if it was, because catch weights were increasing. Nevertheless, our study shows that while the two processes were taking place in the reserve, the benefits differed among fishermen and fish families, which needs to be taken into consideration in future management plans. 

Line 429 – 432: states that “ Our study shows that surveys of recreational fishing activities can constitute robust alternatives for estimating fishery indicators (e.g. CPUE) compared with using data from commercial fisheries …” , This statement should be rephrased to more accurately represent what was achieved. As no comparison is made with either commercial data or fisheries independent data, it is not possible to assess robustness, but the paper does demonstrate the use of a worthwhile alternate approach.

The term “robust” has been replaced by “effective”.

Lines 441 – 442: The terms social and ecological winners appear somewhat out of the blue here (although social and ecological protagonists are mentioned in the introduction). Defining these concepts and their implications in the introduction would strengthen this conclusion when it appears.

The notion of winners/losers is now introduced in the introduction section l. 74-76.

---

## [Editor Report · Decision Letter 1]

17 Nov 2020

Marine reserve benefits and recreational fishing yields: The winners and the losers

PONE-D-20-23582R1

Dear Dr. Kayal,

We’re pleased to inform you that your manuscript has been judged scientifically suitable for publication and will be formally accepted for publication once it meets all outstanding technical requirements.

Kind regards,

Andrea Belgrano, Ph.D.

Academic Editor

PLOS ONE

Additional Editor Comments (optional):

Thank you for addressing in the revised manuscript all the comments/suggestions made during the review process, in particular also the data availability used in this study.

---

## [Editor Report · Acceptance letter]

19 Nov 2020

PONE-D-20-23582R1 

Marine reserve benefits and recreational fishing yields: The winners and the losers 

Dear Dr. Kayal:

I'm pleased to inform you that your manuscript has been deemed suitable for publication in PLOS ONE. Congratulations! Your manuscript is now with our production department. 

Kind regards, 

on behalf of

Dr. Andrea Belgrano 

Academic Editor

PLOS ONE